# A Simple Scheme for Extraction of Asphaltenes from Asphalt at Room Temperature

Dachuan Sun 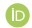

School of Transportation Engineering, Shandong Jianzhu University, Jinan 250101, China; sundc@iccas.ac.cn; Tel.: +86-199-53-120-288

**Abstract:** This paper proposes a simple scheme to separate asphaltenes from asphalt at room temperature without heating or refluxing. The proposed scheme can solve the problems of high energy cost, expensive devices, and safety risks of flammable steam in standard methods of asphaltene extraction. First, the asphalt is dissolved in a good solvent to obtain a solution containing asphaltenes, and the inorganic impurity as well as residual carbons are removed by filtration. Then, the solution containing asphaltenes is dropped into poor solvent to let asphaltenes flocculate into suspended solids. Finally, the suspension is filtered, and the filter cake is dried to obtain asphaltene solid. The CHNS elements and $^1$H-Nuclear magnetic resonance were characterized for the obtained product. Compared with asphalt, the C/H element ratios and the aromatic carbon ratios of the product were higher, which matched the elemental and structural characteristics of asphaltenes. The asphaltene yields obtained from different solvents were compared, and the reasons for the yield differences were analyzed. Recovered solvent could be used to extract asphaltenes, and the yield was found to decrease with the extraction times.

**Keywords:** asphaltene; extraction; asphalt; room temperature

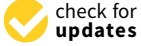



## 1. Introduction

Asphaltene exists in asphalt, heavy oil and residual oil [1] and is the component with the largest relative average molecular weight, strongest polarity, strongest association and highest content of heteroatoms in asphalt [2]. The physical properties of asphalt such as penetration, softening point, ductility, high temperature stability and rutting resistance are great influenced by the content and structures of asphaltene [3]. It is generally believed that asphalt has a colloid structure with asphaltene as the core [4]. To analyze the aggregate structure of asphaltenes and to change the asphaltene content in asphalt, it is necessary to extract asphaltenes from asphalt [5]. To study the physical and chemical properties of asphaltenes, a fast and simple method to extract asphaltenes from asphalt is also required [6]. For preparing polymer-modified asphalt, it is sometimes necessary to modify asphaltenes by chemical means [7]. Hydrogenation and catalytic cracking after selective removal of asphaltenes in heavy oil can avoid catalyst deactivation and coking [8]. Therefore, it is necessary to develop a facile method for extraction of asphaltenes from asphalt at room temperature [9].

Asphaltene is strictly defined as a class of substances in petroleum that are insoluble in lower n-alkanes ($nC_5$–$nC_7$) but soluble in benzene or toluene [10]. Asphaltenes are a class of substances defined in terms of solubility and degree of separation without specific chemical equations [11]. Traditional separation methods such as the asphalt chemical component test (JTG E20-2011 or IP 143) require heating and refluxing, which is cumbersome and time-consuming [12,13]. In addition, specialized equipment is required, and the economic cost is high. As organic solvents such as n-heptane and toluene used for extraction are very flammable, careless operation during heating and refluxing may cause fire or explosion.

In this paper, a simple and economical room temperature separation scheme is used to extract asphaltenes from asphalt without heating or refluxing. Such a physical separation

utilizes the solubility difference of asphalt components and asphaltenes in different solvents. First, asphalt is dissolved in a good solvent to obtain a homogeneous solution, and then it is poured into a poor solvent for asphaltenes, so that the asphaltenes are precipitated in solution and then separated by filtration. This scheme provides a simple and fast method for obtaining asphaltenes. Utilizing the difference of boiling points, the used solvents can be recovered and reused for further extraction, thus reducing the costs of asphaltenes. Guo et al. extracted asphaltenes by directly stirring asphalt with poor solvent for 24 h at room temperature [9]. Such a method cannot remove inorganic impurities and may damage the purity of asphaltenes [9,14,15]. By titrating the poor solvent into a solution of asphaltenes, the onset of flocculation of asphaltenes could be determined. Such a method was used to determine cloud points of phase transition, but it has not yet been used for asphaltene extraction [13,16].

The rest of this paper is organized as follows: the manufacturers of all chemicals and devices used in this study, the experimental procedures and characterization methods are given in the Section 2. Results and discussions related to the experiment process, product characterization, choice of solvent and its effect on asphaltene yield are shown in Section 3. The conclusions and perspectives are in Section 4.

## 2. Materials and Methods

### 2.1. Materials and Devices

The asphalt used is Qilu No. 70 base asphalt from Qilu Petrochemical Company (Zibo, China), which does not contain modifiers or additives such as polymers. Unless otherwise stated, all solvents and other reagents used were of analytical grade, with a purity greater than 99.5% and purchased from Shanghai Macklin Biochemical Technology Company (Shanghai, China). Toluene and xylene (xylene isomer + ethylbenzene, CAS No. 1330-20-7) were used as good solvents for asphaltenes. The poor solvents were straight-chain alkanes such as n-pentane, n-hexane and n-heptane. As shown in Table 1, four groups of experiments were designed utilizing the above good and poor solvents. The first group used xylene as good solvent and n-heptane as poor solvent.

**Table 1.** The solvents used in four groups of experiments.

| Experiment No. | Good Solvent | Poor Solvent |
|:---:|:---:|:---:|
| 1 | Xylene | n-Heptane |
| 2 | Toluene | n-Heptane |
| 3 | Xylene | n-Hexane |
| 4 | Xylene | n-Pentane |

The HJ-6A thermostatic magnetic stirrer was purchased from Jiangsu Zhongda Technology Instrument Company (Nanjing, China). The RE-52AA rotary evaporator and SHZ-III circulating water vacuum pump were purchased from Shanghai Yarong Biochemical Instrument Factory (Shanghai, China). The filter device with volume 1000 mL was purchased from Jiangsu Feida Glass Products Company (Nanjing, China). The medium-speed setting filter paper was purchased from Hangzhou Fuyang Northwood Pulp Company (Hangzhou, China). The 0.45 μm filter membrane was purchased from Shanghai Xinya Purification Device Factory (Shanghai, China). The 200 mL round-bottomed flask was purchased from Chongqing Synthware Glass Company (Chongqing, China). The PX224ZH/E electronic balance and STX portable balance were purchased from Ohaus International Trading (Shanghai) Company (Shanghai, China). Other consumables were supplied by Shandong Laboratory Experimental Instrument Company (Jinan, China).

### 2.2. Experimental Procedures

The flowchart of the experimental process for asphaltene extraction at room temperature is shown in Figure 1. Real pictures for the main procedures are shown in Figure 2.

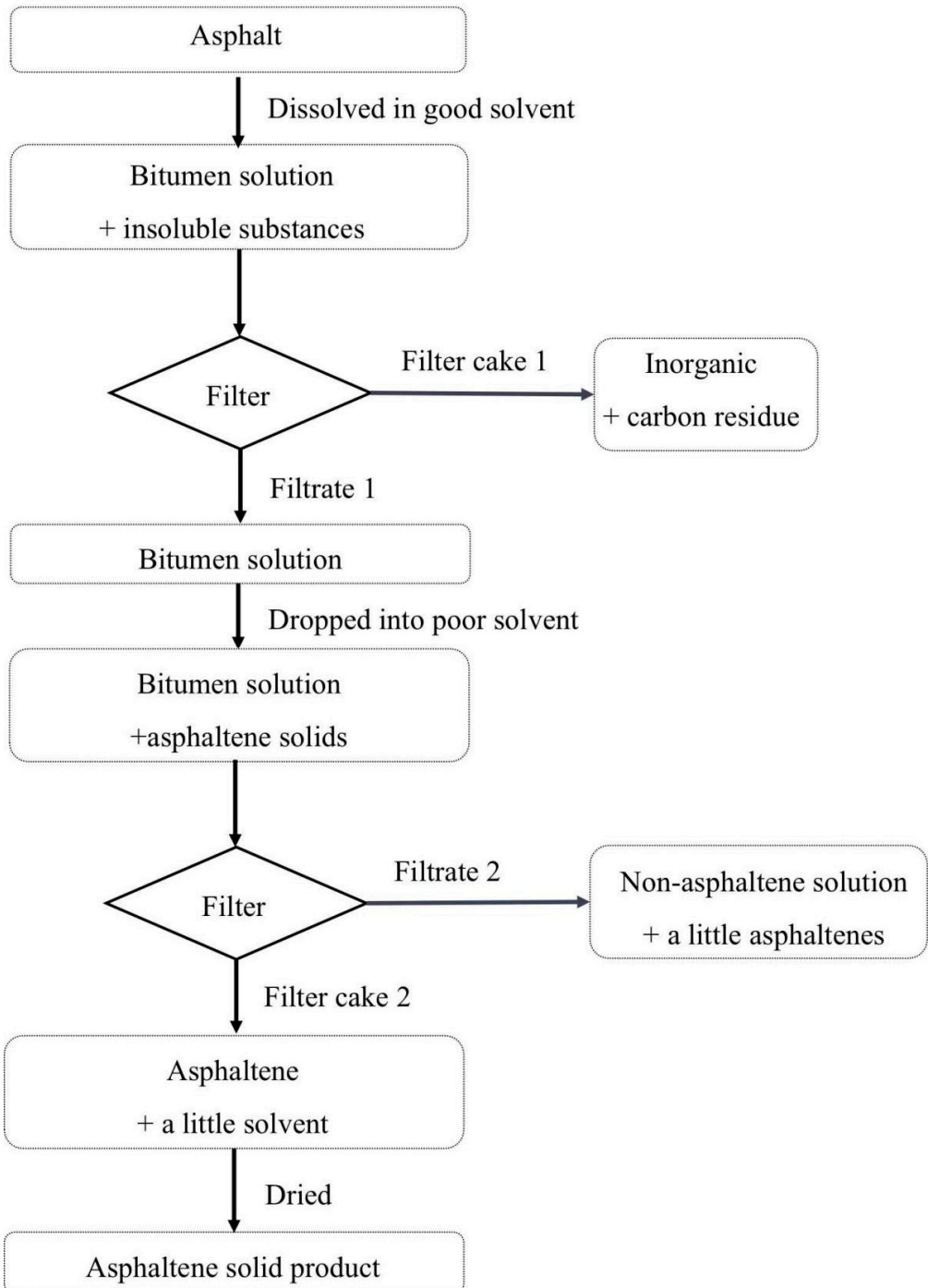

**Figure 1.** Flowchart for extraction of asphaltenes from bitumen at room temperature.

Taking the first group of experiments in Table 1 as an example, the specific experimental steps are:

(1) A total of 5 g asphalt was weighed on a portable balance and then put into a 200 mL round-bottomed flask, as shown in Figure 2a.

(2) After a PTFE coated magnetic stirrer bar with length 2.5 cm was added and 45 mL of xylene was poured into the flask, the solution was stirred vigorously for 30 min to dissolve the asphalt completely on a thermostatic magnetic stirrer, as shown in Figure 2b.

(3) The solution after stirring was filtered to remove inorganic impurities and residual carbon (filter cake 1), as shown in Figure 2c. The filter membrane used was medium-

speed setting filter paper. 5 mL of xylene was used to rinse the filter paper and combined with the filtrate 1. After filtering, the filtrate 1 was set aside for later use.

(4) A total of 450 mL of n-heptane was poured into a 500 mL beaker with magnetic stirring. The above filtrate 1 was dropped into the beaker with a speed of 1–3 drops per second, as shown in Figure 2d.

(5) After dropping, the solution in the beaker was left to stand for 30 min, as shown in Figure 2e.

(6) The solution in the beaker was filtered to get filter cake 2 and filtrate 2, using a filter device with 0.45 um filter membrane, as shown in Figure 2f. The obtained filter cake 2 was washed with 50 mL of n-heptane. The washed solution was combined with the filtrate 2 and set aside for later use.

(7) After filtering and washing, the filter cake 2 was obtained, as shown in Figure 2g. As it still contained a little solvent, it was dried at room temperature for 8 h to constant weight.

(8) The asphaltene after drying was weighed on a PX224ZH/E electronic balance to calculate the yield and then sealed in a 20 mL glass bottle for further characterization, as shown in Figure 2h. The yield was defined as mass percentage of the obtained asphaltenes and matrix asphalt.

(9) If needed, the solvents in filtrate 2 could be recovered by a reduced pressure rotary evaporation, as shown in Figure 2i. The 30–45 °C fraction was obtained as n-heptane 2, and the 60–80 °C fraction was obtained as xylene 2. The volumes for recovered solvents were measured to calculate the recovery rate (volume ratio between the recovered and the added solvents). If solvent recovery is not required, this step can be omitted.

The recovered n-heptane 2 and xylene 2 were used to extract the asphaltenes for the second time, and the yield of the asphaltenes 2 was calculated according to the above operation. The rotary evaporation was used again to recover n-heptane 3 and xylene 3. The recovered n-heptane 3 and xylene 3 were used to extract the asphaltenes for the third time, and the yield of the asphaltenes 3 was calculated. The yields of asphaltenes obtained from the above three recoveries were recorded.

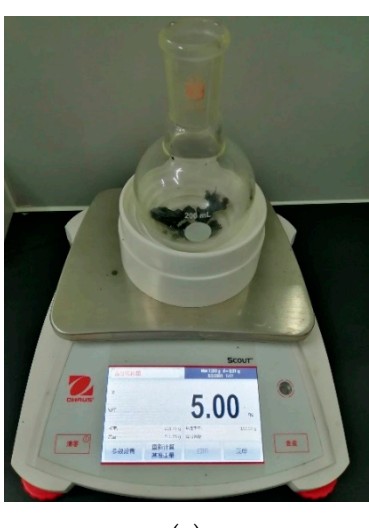
(**a**)

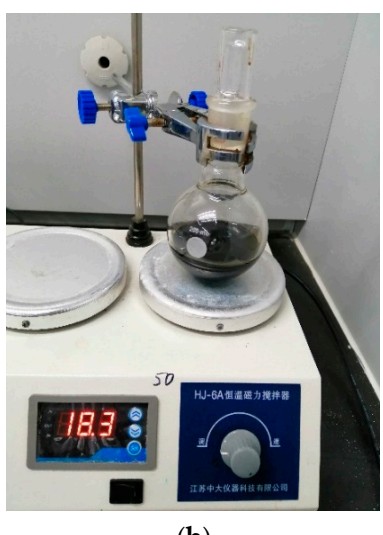
(**b**)

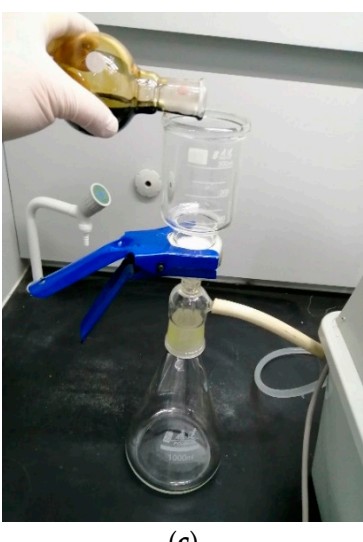
(**c**)

**Figure 2.** *Cont.*

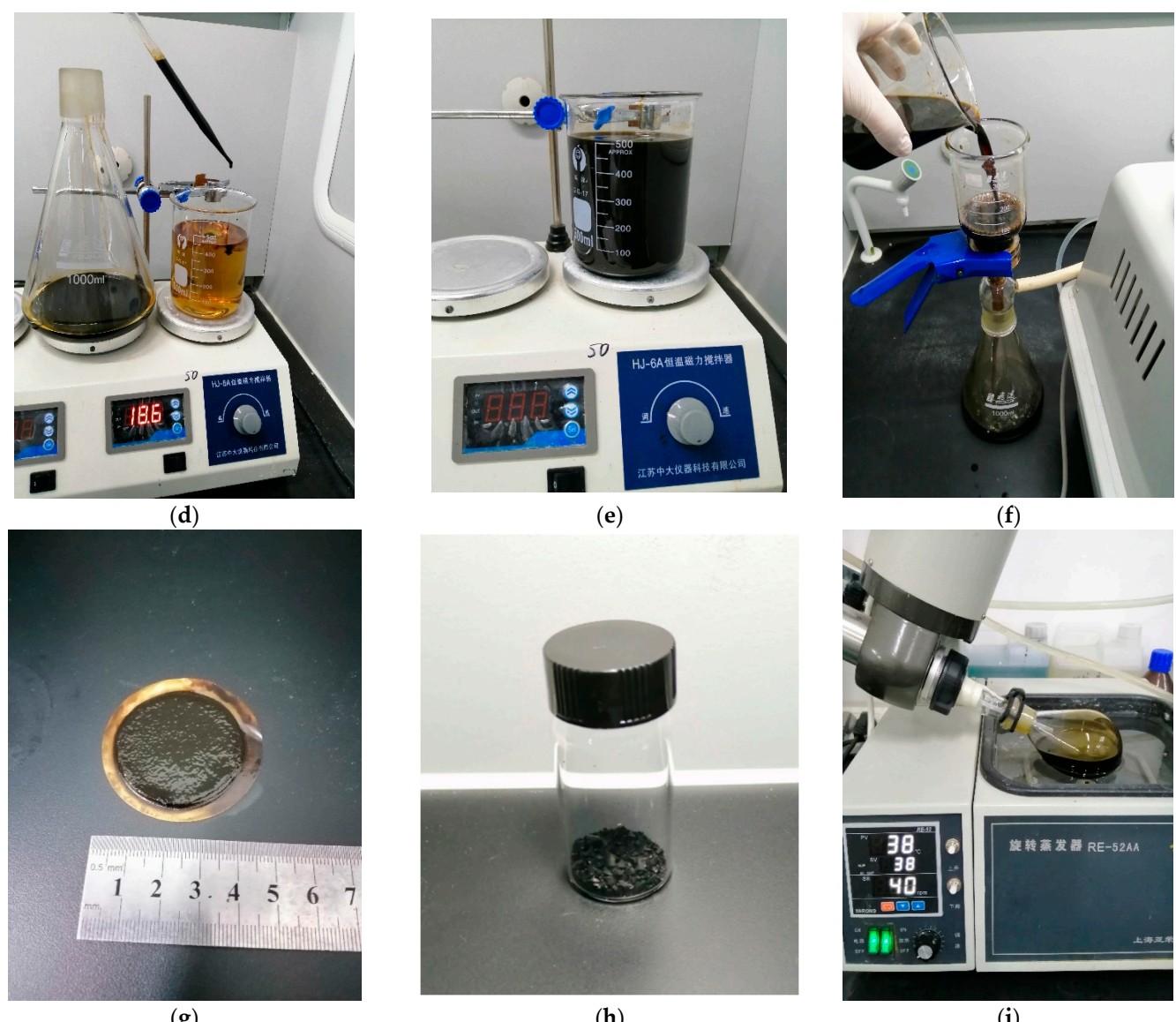

**Figure 2.** Extraction of asphaltenes from bitumen at room temperature: (**a**) 5 g of asphalt in a 200 mL round-bottomed flask; (**b**) asphalt dissolved in xylene on a thermostatic magnetic stirrer; (**c**) the solution after stirring was filtered to get filter cake 1 and filtrate 1; (**d**) the filtrate 1 dropped into the beaker with 450 mL of n-heptane; (**e**) after dropping, the solution in the beaker was left to stand for 30 min; (**f**) the solution in the beaker was filtered to get filter cake 2 and filtrate 2; (**g**) the filter cake 2 after filtering and washing; (**h**) the asphaltene after drying was sealed in a 20 mL glass bottle; (**i**) the solvents in filtrate 2 were recovered by a reduced pressure rotary evaporation.

The second, third, and fourth groups of experiments had the same operation procedures, only replacing the corresponding solvents according to Table 1. For each experiment group or extraction using recovered solvents, five replicates were tested to check the repeatability. The averaged asphaltene yield was calculated for each group, and the difference between these yields did not exceed 0.8%. All experiments were carried out at atmospheric pressure. The room temperature was maintained at 25 °C by an air conditioner. All operations were carried out in a fume hood.

### 2.3. Characterization Methods

The CHNS elemental analysis of the base asphalt and the extracted asphaltenes was performed using the Elimonta UNICUBE organic Elemental Analyzer (EA) (German El-

ements Company China Branch, Shanghai, China). The molecular structures of the extracted asphaltenes and matrix asphalt were characterized by Bruker 600 MHz [1]H-Nuclear magnetic resonance spectroscopy ([1]H-NMR) (Bruker Company, Jinan, China), and the absorption peak areas of each component were integrated by MestReNova software (version: 6.1.0-6224 for Windows).

## 3. Results and Discussion

### 3.1. Experiment Process

Most crude oils or asphalts are composed of stable colloids with asphaltene as the core of the dispersed phase and resins as the solvation layer [17]. Dilution with a large amount of low-molecular-weight n-alkanes can reduce the aromaticity and viscosity of the dispersion system and destroy the colloidal stability, so that the asphaltenes flocculate and precipitate into a separate phase [18]. To solve problems in previous schemes such as heating required, safety risks and time consumed, etc., this paper designs a simple scheme to extract asphaltenes from asphalt at room temperature, using the solubility difference of asphaltenes in different solvents.

As shown in Figure 1, the asphalt is dissolved in a solvent to obtain a solution containing asphaltenes at first, and the inorganic substances and residual carbon are removed by filtration. Then, the abovementioned solution containing asphaltenes is dropped into n-alkanes to precipitate asphaltenes and obtain suspensions. Finally, the above suspension is allowed to stand and then filtered, and the filter cake is dried to obtain asphaltene solids. When the standing time was in the range of 30 min to 24 h, no significant change in the yield was found, so standing for 30 min was considered to be sufficient. The drying time usually needs only 4 h at room temperature to achieve constant weight, and if vacuum drying at room temperature is used, it only takes 2 h to achieve constant weight. The room temperature mentioned in this paper is generally defined as 25 °C, but this scheme can still be used to extract asphaltenes within the range of 10 °C above room temperature or 30 °C below room temperature.

As shown in Figure 2h, the asphaltenes extracted from asphalt with different solvents are all brittle solids with coal-like luster. No apparent adhesion to glass, skin or plastics was found at room temperature. Therefore, the relation between asphaltene content with the adhesion strength and coating ability of asphalt to aggregates may need further studying.

One advantage of this scheme is that it saves time and money. Asphaltene can be separated and extracted at room temperature, avoiding the economic cost of purchasing specialized equipment such as reflux devices, electric heating plates or heating jackets in traditional methods as well as energy costs during the heating and refluxing process. As no heating and cooling processes are required, this method consumes less time compared to traditional methods. Another advantage is the ability to avoid safety hazards caused by heating. Since the n-alkanes used (such as n-heptane) have a low flash point, the organic vapor generated during refluxing may leak and easily catch fire, which constitutes safety hazards to the laboratory and operators.

### 3.2. Characterization of Extracted Asphaltenes

Elemental and chemical structural characterizations were performed on the asphaltenes extracted from the first group of experiments as well as on the matrix asphalt. The experimental results of elemental analysis in Table 2 were obtained using an organic elemental analyzer. C and H elements had the largest fractions, with a total amount around 90%, consistent with previous experimental results [9]. C/H atomic ratio is defined as $q$. Cimino et al. analyzed the mass percentages of CHN elements for extracted asphaltenes from refluxing and found that C/H mass ratio was as high as 13.38 for $nC_5$ and 14.21 for $nC_7$ solvents [16]. According to Table 2, the C/H mass ratio of the extracted asphaltene was 11.34 ($q$ = 0.94), which is higher than the C/H mass ratio of the base asphalt of 9.40 ($q$ = 0.78), so the unsaturation degree of the extracted asphaltene is higher than the base asphalt. The $q$ values calculated according to Table 2 are listed in Table 3. Compared with the base asphalt, the

content of C and H elements in the extracted asphaltene decreased, while the ratio of N and S elements increased. The N and S elements could introduce polar groups into the asphaltene molecules, increasing their molecular polarity, the aggregation ability, and the coating ability [19].

**Table 2.** Mass percentage of elements from elemental analysis.

| Fractions | N (%) | C (%) | H (%) | S (%) | Others (%) |
|---|---|---|---|---|---|
| Asphalt | 0.74 ± 0.07 | 83.09 ± 0.12 | 8.83 ± 0.05 | 4.19 ± 0.07 | 3.14 ± 0.31 |
| Asphaltene | 1.55 ± 0.06 | 82.33 ± 0.12 | 7.25 ± 0.06 | 5.97 ± 0.06 | 2.88 ± 0.30 |

**Table 3.** Structural parameters of asphaltenes and base asphalt [1].

| Structural Parameters | Asphaltene | Asphalt |
|---|---|---|
| $q$ | 0.94 | 0.78 |
| $h_A$ | 0.12 | 0.04 |
| $h_\alpha$ | 0.22 | 0.15 |
| $h_\beta$ | 0.50 | 0.63 |
| $h_\gamma$ | 0.16 | 0.17 |
| $f_A$ | 0.53 | 0.39 |

[1.] $q$ is the C/H atomic ratio; $h_A, h_\alpha, h_\beta, h_\gamma$ are respectively the hydrogen atomic fractions of aromatic hydrocarbons, $\alpha$ aromatic hydrocarbons, $\beta$ aromatic hydrocarbons, and $\gamma$ aromatic hydrocarbons; $f_A$ is aromatic carbon ratio.

The molecular structures of the extracted asphaltenes, as well as the base asphalt, are characterized by NMR in Figure 3. The hydrogen atomic fractions calculated from Figure 3 are listed in Table 3. The aromatic carbon ratio $f_A$ was calculated according to: [20]

$$f_A = (q - (h_\alpha + h_\beta + h_\gamma)/2)/q, \tag{1}$$

in which, $h_A, h_\alpha, h_\beta, h_\gamma$ are respectively the hydrogen atomic fractions of aromatic hydrocarbons $H_A$; $\alpha$ aromatic hydrocarbons $H_\alpha$; $\beta$ aromatic hydrocarbons $H_\beta$, and $\gamma$ aromatic hydrocarbons $H_\gamma$.

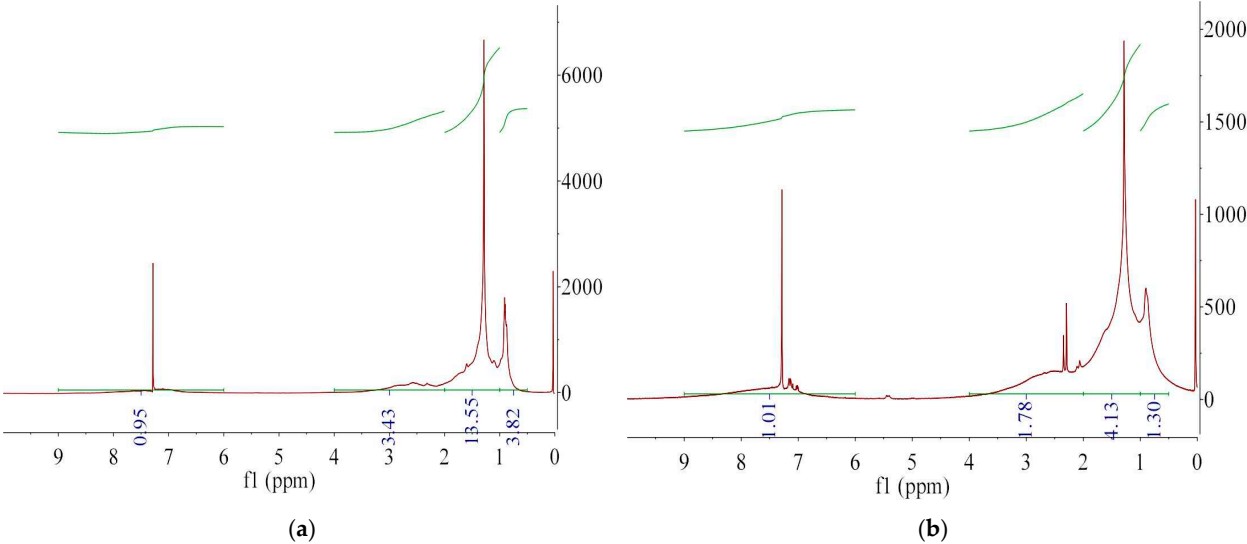

**Figure 3.** The molecular structures of the extracted asphaltenes and the base asphalt characterized by NMR: (**a**) Hydrogen NMR spectrum of base asphalt; (**b**) NMR spectrum of the obtained asphaltenes. The hydrogen content was integrated using MestReNova software to calculate the proportion of hydrogen atoms in different categories and the structural parameters.

As shown in Figure 4, the hydrogen atoms of petroleum samples are divided into four categories [9,20]: aromatic carbon hydrogens $H_A$ (hydrogen directly attached to aromatic carbons, chemical shift 6.0–9.0 ppm); $\alpha$ aromatic carbon hydrogens $H_\alpha$ (hydrogen attached to $\alpha$ carbons of aromatic rings, chemical shift 2.0–4.0 ppm); $\beta$ aromatic carbon hydrogens $H_\beta$ (hydrogen connected to $\beta$ carbon of aromatic ring and hydrogen on $CH_2$ and CH farther from $\beta$, chemical shift 1.0–2.0 ppm) and $\gamma$ aromatic carbon hydrogens $H_\gamma$ (hydrogen attached to the $\gamma$ position and the hydrogen on $CH_3$ farther from the $\gamma$ position, chemical shift 0.5–1.0 ppm). MestReNova software was used to integrate the absorption peak areas in each component, and the hydrogen atomic fractions for the above four categories were calculated, as shown in Table 3. Combining this data with a modified Brown–Ladner method, the structures of asphaltenes and their subcomponents can be inferred.

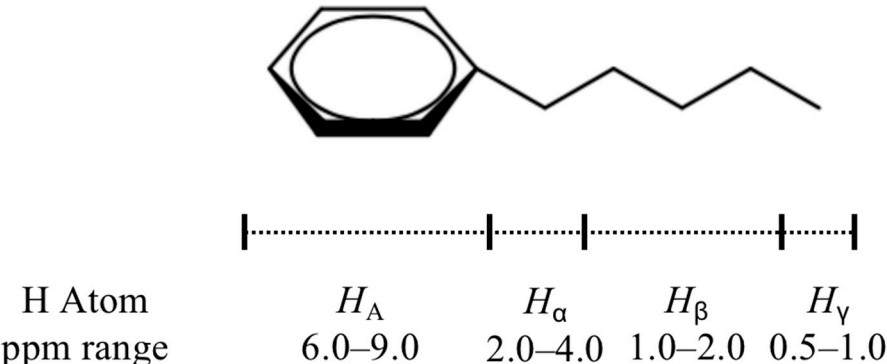

**Figure 4.** Hydrogen atoms are divided into four categories and the corresponding ppm range on NMR spectroscopy for each category.

Table 3 shows that the atomic fraction of aromatic carbon hydrogens $H_A$ contained in the extracted asphaltenes is $h_A = 0.12$, which is higher than that of the asphalt 0.04, indicating that the asphaltenes have more aromatic structures and are more unsaturated than the asphalt. This is consistent with the C/H mass ratio results in Table 2. For both asphaltene and base asphalt, the proportion of $H_A$ hydrogen atoms is relatively small, and the highest proportion is $\beta$ carbon hydrogens $H_\beta$. The aromatic carbon ratio $f_A$ of asphaltenes is around 0.50, indicating that the aromatic ring structure in the molecular structure is well developed, consistent with the previous reports [20,21]. Asphaltenes have higher $h_A$ and $h_\alpha$ and smaller $h_\beta$ than asphalt, indicating that asphaltenes have larger aromatic rings, more carbon chains connected with the aromatic ring but shorter side chains than asphalt, consistent with the previous reports [9,22]. The above results are consistent with the elemental and structural characteristics of asphaltenes, which proves that the extracted materials are asphaltenes.

### 3.3. Choice of Solvent and Its Effect on Asphaltene Yield

Solvents that can dissolve asphaltenes include aromatic hydrocarbons, carbon disulfide and some halogen-containing organic solvents [23]. It is generally believed that asphaltenes have archipelagic and continental molecular structures with alkanes and aromatic parts linked by chemical bonds [24]. Asphaltenes have a high degree of unsaturation. Toluene and xylene have similar molecular structures to asphaltenes, so they are used as good solvents in this paper. According to Table 4, the C/H mass ratios of toluene and xylene are 10.51 and 9.61, respectively, and their atomic ratios are 0.88 and 0.80, respectively. Therefore, toluene has a higher degree of unsaturation than xylene and greater solubility for asphaltenes. However, after being dropped into the poor solvent, the residual toluene has a stronger inhibitory effect on the precipitation of asphaltenes, thus causing more asphaltenes to remain in the solution, and resulting in a lower yield of asphaltenes as shown in Table 5.

**Table 4.** Chemical formula, mass and atomic ratios of C/H elements for different solvents.

| Solvents | Chemical Formula | C/H Mass Ratio | C/H Atomic Ratio |
|---|---|---|---|
| Xylene | $C_8H_{10}$ | 9.61 | 0.80 |
| Toluene | $C_7H_8$ | 10.51 | 0.88 |
| n-Pentane | $C_5H_{12}$ | 5.00 | 0.42 |
| n-Hexane | $C_6H_{14}$ | 5.15 | 0.43 |
| n-Heptane | $C_7H_{16}$ | 5.25 | 0.44 |

**Table 5.** Asphaltene yields in different good and poor solvents.

| Experiment No. | Good Solvent | Poor Solvent | Asphaltene Yield |
|---|---|---|---|
| 1 | Xylene | n-Heptane | $13.2 \pm 0.4\%$ |
| 2 | Toluene | n-Heptane | $11.3 \pm 0.4\%$ |
| 3 | Xylene | n-Hexane | $13.7 \pm 0.4\%$ |
| 4 | Xylene | n-Pentane | $16.8 \pm 0.4\%$ |

When the poor solvents are n-heptane, the yield of asphaltenes with xylene is $13.2 \pm 0.4\%$ while that with toluene is $11.3 \pm 0.4\%$. Guo et al. also extracted asphaltenes at room temperature by using n-heptane as poor solvent without using good solvent [9]. Their asphaltene yield was $17.5 \pm 1.5\%$, higher than the yields in Table 5. According to Sakib and Bhasin's results [15], the yield of extracted asphaltenes at room temperature was in range of 15.9–26.6%, which was also higher than yields in Table 5. These higher yields may arise from differences in the source of asphalt or from absence of good solvent. As good solvent was not used, the solubility for asphaltenes would be reduced and the yield should rise.

The poor solvent should have little solubility to asphaltenes. In this paper, three organic solvents, n-pentane, n-hexane and n-heptane, are used as poor solvents. According to Table 4, from n-pentane to n-heptane, the C/H mass ratio and the C/H atomic number ratio increase sequentially. The discrepancies decrease between these solvents and asphaltenes, as asphaltenes have C/H mass ratio 11.34 and the C/H atomic number ratio 0.94. Therefore, the ability to flocculate asphaltenes increases with the carbon chain length. When all good solvents are xylene, as in Table 5, the yields of asphaltenes with alkane carbon numbers of five to seven decrease from $16.8 \pm 0.4\%$ to $13.2 \pm 0.4\%$. Previous studies have also found that the precipitation ability of n-pentane to asphaltenes is stronger than that of n-heptane [25]. Some asphaltenes can be dissolved in n-heptane but not in n-pentane. This part of asphaltenes has a lower molecular weight and a higher degree of saturation, does not form a stacking structure, and its molecular structure is closer to resins [25].

Compared with toluene, the yield of xylene is higher, and xylene is cheap and not a controlled solvent, so it is more suitable to use as a good solvent for extracting asphaltenes. Compared with n-pentane and n-hexane, n-heptane has a higher boiling point and is easier to recover. We therefore recovered the solvent from the first group of experiments and used it directly for the re-extraction of asphaltenes. The yields are listed in Table 6. A high asphaltene yield of $10.8 \pm 0.4\%$ can still be obtained by utilizing the recovered solvent in the two-time extraction. Thus, the solvents used in previous extractions can be recycled to a certain extent, reducing the extraction cost of asphaltenes. With increasing extraction times, however, the yield of asphaltenes decreased gradually from $13.2 \pm 0.4\%$ to $8.3 \pm 0.4\%$. This is because xylene and n-heptane can form an azeotrope and can be distilled out together. With the increase of extraction times, the purity of n-heptane and xylene decreases. Since the poor solvent n-heptane has more volume, the xylene dissolved in n-heptane can hinder the precipitation of partial asphaltenes and reduce the yield of extracted asphaltenes.

**Table 6.** Yields of extracted asphaltenes using recovered solvents at different extraction times.

| Extraction Times | Good Solvent | Poor Solvent | Asphaltene Yield |
|:---:|:---:|:---:|:---:|
| 1 | Xylene | n-Heptane | $13.2 \pm 0.4\%$ |
| 2 | Xylene 2 | n-Heptane 2 | $10.8 \pm 0.4\%$ |
| 3 | Xylene 3 | n-Heptane 3 | $8.3 \pm 0.4\%$ |

The simple method for extracting asphaltenes from asphalt proposed in this paper can be used for extracting asphaltenes from asphalt in laboratories or industry. It avoids the risk of fire and explosion caused by heating and refluxing. As shown in Table 7, it has the advantages of simple and fast operation, lower energy consumption and more safety, compared to previous methods with heating or refluxing procedures [12,13]. Compared to other extraction methods at room temperature [9,14,15], the method proposed in this study could remove inorganic impurity and residual carbons from asphaltenes by the first filtering in Figure 1, resulting in a purer asphaltene product.

**Table 7.** Performance improvement by comparison between the method proposed in this study and other extraction methods from references. O and × represent yes and no, respectively.

| No Refluxing | Fast and Simple | Inorganic Impurity Removable | References |
|:---:|:---:|:---:|:---:|
| O | O | O | This work |
| O | O | × | [9,14,15] |
| × | × | O | [12,13] |

One drawback of our scheme is that good solvent is needed, and the separation between good and poor solvent is still energy-consuming. Fortunately, the solvent can be recovered and reused to lower partial costs for asphaltenes. To increase asphaltene yield, further research is needed to improve the separation efficiency between good and poor solvents. When the amounts of inorganic impurity and carbon residues in asphalt sources are acceptably small, the use of good solvent could be omitted and asphaltene can be extracted by using poor solvent only. The flowchart would then change from Figure 1 to Figure 5. The proportion of inorganic impurity and carbon residues of the asphalt in this study is only around 0.1% wt. Following the flowchart in Figure 5, 2.5 g asphalt and 250 mL n-heptane were stirred for 1 h and then filtered and dried. The asphaltene yield was $19.7 \pm 0.4\%$, higher than the yields in Tables 5 and 6. As good solvent causes some asphaltenes to remain in the solution, some asphaltenes remain rather than flocculate, thereby reducing the yield. For other sources of asphalt, a procedure following the flowchart in Figure 1 is suggested, unless the amount of the inorganic impurity and residual carbons are identified and sufficiently small.

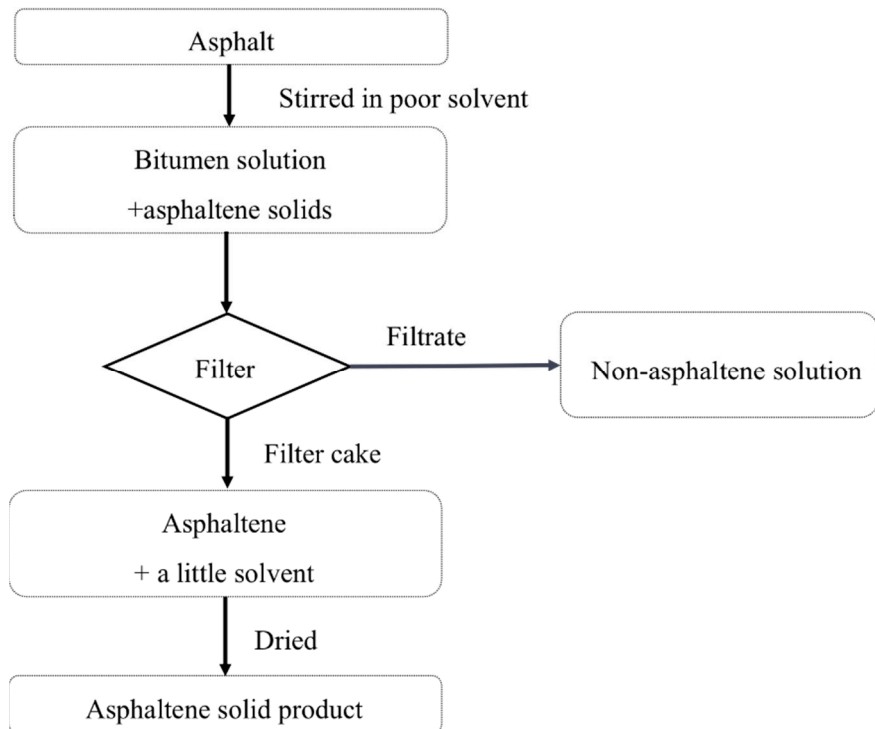

**Figure 5.** Flowchart for extraction of asphaltenes from bitumen at room temperature, when the amounts of inorganic impurity and carbon residues in asphalt sources are small enough.

## 4. Conclusions

According to the solubility difference of asphaltenes in different solvents, a scheme for extracting asphaltenes at room temperature was proposed. Elemental and chemical structural characterizations were performed on the asphaltenes extracted from this scheme. Effects of different good and poor solvents on the extraction yields of asphaltenes were studied. According to organic element analysis, the C/H ratio of the asphaltenes was 11.34, which was significantly higher than that of the base asphalt 9.40. Compared with the base asphalt, the content of C and H elements in the extracted asphaltenes decreased, while the proportion of N and S elements increased. $^1$H-NMR results of aromatic carbon hydrogens, α, β and γ aromatic carbon hydrogens indicated that the extracted asphaltenes had larger aromatic rings, more carbon chains connected with the aromatic ring but shorter side chains than the base asphalt. The solvents could be recovered by rotary evaporation and further reused to extract asphaltenes. However, the yield of asphaltenes decreased with extraction times. Compared to standard methods of asphaltene extraction, this method has the advantages of simple and fast operation, lower energy consumption and more safety. Compared to other methods for extracting asphaltenes at room temperature, this method can remove the inorganic impurity and residual carbons. If the amount of inorganic impurity and residual carbons is negligible, asphaltene can be extracted with only poor solvent without good solvent.

The future research plan is to check the feasibility of the simple and fast method proposed in this study in extraction of asphaltenes from different sources of asphalts. Moreover, the adhesion of the remaining asphalt to aggregates after asphaltene extraction needs to be studied to illustrate the effects of asphaltene contents on the adhesion of asphalt to material interfaces.

**Funding:** This research received no external funding.

**Institutional Review Board Statement:** Not applicable.

**Informed Consent Statement:** Not applicable.

**Data Availability Statement:** The data presented in this study are available on request from the corresponding author. The data are not publicly available due to privacy.

**Acknowledgments:** The author acknowledges Ruibo Ren and Pinhui Zhao for their donation of asphalt used for experiments.

**Conflicts of Interest:** The author declares no conflict of interest.

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
