# Peer review of "A Simple Scheme for Extraction of Asphaltenes from Asphalt at Room Temperature"

_coatings, doi:10.3390/coatings12030407_

Round 1

Reviewer 1 Report

The authors describe the experimental investigation of a simple scheme to obtain asphaltenes from asphalt at room temperature without heating or refluxing processes. In summary, the manuscript is well organized and written logically; however, this manuscript can be considered for publication after considerable improvements based on the following comments and suggestions:

  1. In the Introduction section, the authors should add more literature related to the various separation methods that can be used at room temperature.

  1. The authors provided the experimental procedures for extraction of asphaltenes; however, for clear and easy understanding, could you please add a schematic diagram with real pictures of the results step-by-step?

  1. In the Materials and Methods section, manufacturer of all chemicals and devices used in this study should be mentioned.

  1. There is no information on the repeatability of the separation method proposed in this study. In addition, plots with uncertainty can help us understand and compare results.

  1. It is difficult to completely understand the Figure 2 because the resolution of it is very low and the explanation about this figure is not sufficient.

  1. Could you please provide us any comparison to other experimental results at room temperature?

  1. Please provide more descriptions and schematics to understand the mechanism by which the performance is improved by the method proposed in this study.

Author Response

Dear Reviewer,

     Thank you for reviewing my manuscript and thanks for your suggestion and comments. Below is my answers (labelled in blue color):

In the Introduction section, the authors should add more literature related to the various separation methods that can be used at room temperature.

Answer: Yes, literatures with separation methods used at room temperature are added in the “1. Introduction” part in the revised manuscript, such as Ref. 14 and 15.

The authors provided the experimental procedures for extraction of asphaltenes; however, for clear and easy understanding, could you please add a schematic diagram with real pictures of the results step-by-step?

Answer: Yes, in the newly revised manuscript, Figure 1 is the scheme for experimental procedure and Figure 2 contains real pictures for each step. 

In the Materials and Methods section, manufacturer of all chemicals and devices used in this study should be mentioned.

Answer: Yes, manufacturer of all chemicals and devices used in experiments are described in the revised “2.1. Materials and devices” part.

There is no information on the repeatability of the separation method proposed in this study. In addition, plots with uncertainty can help us understand and compare results.

Answer: Yes, the information on the repeatability of the separation method proposed in this study is added in the revised manuscript near line 157:

It is difficult to completely understand the Figure 2 because the resolution of it is very low and the explanation about this figure is not sufficient.

Answer: Yes, the NMR plots in Figure 2 are redrawn in revised paper and labelled as Figure 3 now. More explanation about this figure is added in the text. A new Figure 4 is added to illustrate that hydrogen atoms are divided into 4 categories and the corresponding ppm range on NMR spectroscopy for each category.  

Could you please provide us any comparison to other experimental results at room temperature?

Answer: Yes, several references with experiments done at room temperatures are included in the revised manuscript and the comparison is made between experimental results from these references and our results in the “3. Results and Discussion” part, for example, near line 210, 213, 260, 263, 283, 284, 299.

Please provide more descriptions and schematics to understand the mechanism by which the performance is improved by the method proposed in this study.

Answer: Yes, a new Table 7 is added in the revised manuscript to help understand the mechanism by which the performance is improved by the proposed method in this study.

Best wishes,

Dachuan Sun

Reviewer 2 Report

The topic of the paper is interesting and fits the scopes of the Journal. Nevertheless, the manuscript requires some extra efforts to improve its quality and presentation. After a careful revision, a set of comments are given below.

The term “Asphalt” could be added as keyword, if the authors agree.

In the first section, a common practice in scientific papers consists on including a paragraph at the end of the Introduction to briefly describe the structure of the rest of sections in the manuscript. It is suggested to add such a paragraph for a better readability.

The text between lines 70 and 79 should be formatted with bullet style since it is a list of steps. Another option, more interesting for a high-quality paper consists on adding a flowchart which depicts the described steps in a graphical manner, more illustrative for the reader than solely the text. In fact, the flowchart of figure 1 is very similar.

With respect to the software MestreNova, the particular version that has been used should be mentioned. In addition, the operating system could also be mentioned since this software, as the author knows, is supported by Windows, Linux and Mac. This information can be useful for the interested reader.

Concerning figure 1, the associated text should be placed before such a figure. Indeed, starting a subsection directly with a figure does not contribute to a proper reading.

The indication of Bruker equipment in line is equal to that found in previous line 101. Hence, the author should avoid repeating the abbreviation NMR.

A photograph of the experimental framework would enrich the paper. It could be placed in the second section.

Equation 1 is included in the text with a statement, line 169, which should be enhanced. For example “The aromatic carbon ratio was calculated according to equation (1) from [15]:” or a similar form.

In figure 2, the letter size is too small, making difficult to read the numbers and interpret the graphs.

The Conclusions section should be enlarged. For instance, the main limitations of the work should be commented in a brief manner for a better presentation. In addition, future research guidelines that the authors are considering on the view of the achieved results should be mentioned.

Moreover, the novelties and contribution of the reported research to the body of knowledge needs to be highlighted in this section or, alternatively, in a new section devoted to discuss the achieved results. If this last option is chosen, the previously mentioned limitations of the work, should be placed within this discussion section.

Author Response

Dear Reviewer,

     Thank you for reviewing my manuscript and thanks for your suggestion and comments. Below is my answers (labelled in blue color):

The term “Asphalt” could be added as keyword, if the authors agree.

Answer: Yes, “Asphalt” is added into keywords in the revised manuscript.

In the first section, a common practice in scientific papers consists on including a paragraph at the end of the Introduction to briefly describe the structure of the rest of sections in the manuscript. It is suggested to add such a paragraph for a better readability.

Answer: Yes, such a paragraph is added in the revised manuscript, at the end of the Introduction part. 

The text between lines 70 and 79 should be formatted with bullet style since it is a list of steps. Another option, more interesting for a high-quality paper consists on adding a flowchart which depicts the described steps in a graphical manner, more illustrative for the reader than solely the text. In fact, the flowchart of figure 1 is very similar.

Answer: Yes, the experimental procedure is rewritten as a list of steps (1)~(9) in “2.2 Experimental procedures” part. A new Figure 2 is included with real pictures for each step, to make the experimental procedure more illustrative for the reader.

With respect to the software MestreNova, the particular version that has been used should be mentioned. In addition, the operating system could also be mentioned since this software, as the author knows, is supported by Windows, Linux and Mac. This information can be useful for the interested reader.

Answer: Yes, the version of the software MestreNova and the operating system is added into the last sentence of “2.3. Characterization methods” part, as “version: 6.1.0-6224 for Windows”.

Concerning figure 1, the associated text should be placed before such a figure. Indeed, starting a subsection directly with a figure does not contribute to a proper reading.

Answer: Yes, the figure 1 is moved behind the associated text in the revised manuscript.

The indication of Bruker equipment in line is equal to that found in previous line 101. Hence, the author should avoid repeating the abbreviation NMR.

Answer: Yes, the repetition is solved in the revised manuscrip. The abbreviation NMR firstly appears in the last sentence of “2.3. Characterization methods” part, line 165.

A photograph of the experimental framework would enrich the paper. It could be placed in the second section.

Answer: Yes, the revised Figure 2 contains real pictures for each step of experimental procedure in the “2.2 Experimental procedures” part.

Equation 1 is included in the text with a statement, line 169, which should be enhanced. For example “The aromatic carbon ratio was calculated according to equation (1) from [15]:” or a similar form.

Answer: Yes, the statement for equation 1 is enhanced and physical variables in equation 1 are explained in the revised manuscript.

In figure 2, the letter size is too small, making difficult to read the numbers and interpret the graphs.

Answer: Yes, the letter size is enlarged in the revised manuscipt.

The Conclusions section should be enlarged. For instance, the main limitations of the work should be commented in a brief manner for a better presentation. In addition, future research guidelines that the authors are considering on the view of the achieved results should be mentioned.

Answer: Yes, the main limitations and future research perspectives are included in the revised manuscript.

Moreover, the novelties and contribution of the reported research to the body of knowledge needs to be highlighted in this section or, alternatively, in a new section devoted to discuss the achieved results. If this last option is chosen, the previously mentioned limitations of the work, should be placed within this discussion section.

Answer: Yes, the novelties and contribution of the reported research to the body of knowledge is presented in “conclusion” part, near line 363.

Best wishes,

Dachuan Sun

Reviewer 3 Report

The work by Sun presents a simple method for extraction of asphaltenes from asphalt. The topic is interesting. However, there is a main concerning, it is not clear the impact of the work for coating technology. Some additional comments:

-The material and methods section should be divided in chemicals, procedures and characterization. In the present form, it appears very difficult to follow.

-Figures should be cited in the text before its introduction.

-The concept of micelles in the text is not clear. Micelles involve an association colloids characterized by the existence of two zones with well differentiated hydrophobicities.

Author Response

Dear Reviewer,

     Thank you for reviewing my manuscript and thanks for your suggestion and comments. Below is my answers (in blue color):

-The material and methods section should be divided in chemicals, procedures and characterization. In the present form, it appears very difficult to follow.

Answer: Yes, the “2. Materials and Methods” part is divided into 3 part, including chemicals, procedures and characterization in the revised manuscript.

-Figures should be cited in the text before its introduction.

Answer: Yes, the manuscript is revised and checked, to make sure that each figure is cited in the text before its introduction in the revised manuscript.

-The concept of micelles in the text is not clear. Micelles involve an association colloids characterized by the existence of two zones with well differentiated hydrophobicities.

Answer: Yes, colloid is a more accurate statement than micelle. Thus the micelles in the text are replaced by colloids in the revised manuscript.

Best wishes,

Dachuan Sun

Round 2

Reviewer 3 Report

Authors have addressed all my questions, and now the article is publishable